

# Scattering description of Andreev molecules

**Jean-Damien Pillet[1,2], Vincent Benzoni[1], Joël Griesmar[1],
Jean-Loup Smirr[1] and Çağlar Ö. Girit[1⋆]**

**1** $\Phi_0$, JEIP, USR 3573 CNRS, Collège de France, PSL Research University,
11, place Marcelin Berthelot, 75231 Paris Cedex 05, France
**2** LSI, Ecole Polytechnique, CEA/DRF/IRAMIS, CNRS, Institut Polytechnique de Paris,
F-91128 Palaiseau, France

⋆ caglar.girit@college-de-france.fr

## Abstract

An Andreev molecule is a system of closely spaced superconducting weak links accommodating overlapping Andreev Bound States. Recent theoretical proposals have considered one-dimensional Andreev molecules with a single conduction channel. Here we apply the scattering formalism and extend the analysis to multiple conduction channels, a situation encountered in epitaxial superconductor/semiconductor weak links. We obtain the multi-channel bound state energy spectrum and quantify the contribution of the microscopic non-local transport processes leading to the formation of Andreev molecules.

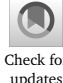
## 1 Introduction

The physical properties of Josephson junctions, both isolated and in ensembles, are well understood and exploited in various fields such as magnetometry and metrology [1]. Due to their

quantum coherence and potential for integration in large-scale circuits, Josephson junctions also serve as superconducting qubits for quantum information and computation [2].

Recently we elucidated an unconventional coupling mechanism, historically referred to as the "order-parameter" interaction [3] or quartets [4], between two closely spaced Josephson junctions [5]. For two weak links separated on the order of the superconducting coherence length $\xi_0$, this coupling arises from the hybridization of quasiparticles to form a molecular, or multi-weak link, Andreev Bound State (ABS). The results were obtained from an analysis of the Bogolubiov-de Gennes (BdG) equations describing an inhomogeneous superconductor in one dimension, with the "Andreev molecule" comprised of two $\delta$-potential weak links separated by a finite superconductor of length $l$. Although the BdG approach is sufficient to develop an intuitive understanding of the phenomenon, it is unwieldy when applied to complicated structures or to weak links with multiple conduction channels.

In an isolated Josephson junction with multiple conduction channels, each channel hosts independent ABS. The total supercurrent is given by the sum of the contributions from each channel, a function of the overall superconducting phase difference and the individual channel transmissions [6]. However, when two multi-channel junctions are placed close to each other, each ABS at the left junction can potentially couple to every ABS at the right junction to form complex Andreev molecules. This situation is relevant since many quantum conductors used in weak links have lateral dimensions comparable to or larger than the Fermi wavelength and thus host multiple channels. For example, in epitaxial superconductor/semiconductor nanowires [7] or 2D electron gases [8], one can readily tune the number of channels with a local gate electrode [9,10].

Here we apply the scattering matrix formalism to describe Andreev molecules with multiple channels. First we formulate the problem for the case of two weak links connected to three superconductors, introducing new terms accounting for partial Andreev reflection at the finite central superconductor. We identify the microscopic scattering processes, elastic cotunneling and crossed Andreev reflection, which give rise to ABS hybridization. After verifying that the results are consistent with the BdG treatment of a single-channel molecule, we calculate the energy spectra of a twenty-channel Andreev molecule. Finally we depict the extended quasiparticle trajectories arising in an Andreev molecule and plot their probabilities as a function of the size of the finite superconductor.

## 2 Scattering Formalism

A convenient approach to treat conduction through mesoscopic systems is the Landauer-Büttiker scattering formalism [11]. Matrices describe the scattering of propagating electrons or holes on three different types of elements: weak links, semi-infinite superconductors, and a superconductor of finite length, Fig. 1(a). In this approach, electrons and holes ($e$ and $h$) are described by an ensemble of waves propagating to the left or the right ($\leftarrow$ or $\rightarrow$), which are connected to each other by normal scattering processes at the left or right ($L$ or $R$) weak link or Andreev processes on the three superconductors.

These waves can be labeled with two sets of eight coefficients

$$
\begin{aligned}
\mathcal{A} &= \left(a_{Le}^{\rightarrow}, a_{Le}^{\leftarrow}, a_{Re}^{\rightarrow}, a_{Re}^{\leftarrow}, a_{Lh}^{\rightarrow}, a_{Lh}^{\leftarrow}, a_{Rh}^{\rightarrow}, a_{Rh}^{\leftarrow}\right)^T, \\
\mathcal{B} &= \left(b_{Le}^{\leftarrow}, b_{Le}^{\rightarrow}, b_{Re}^{\leftarrow}, b_{Re}^{\rightarrow}, b_{Lh}^{\leftarrow}, b_{Lh}^{\rightarrow}, b_{Rh}^{\leftarrow}, b_{Rh}^{\rightarrow}\right)^T,
\end{aligned}
\tag{1}
$$

where $\mathcal{A}$ describes waves propagating towards weak links with amplitudes $a$ and $\mathcal{B}$ describes outgoing waves with amplitudes $b$.

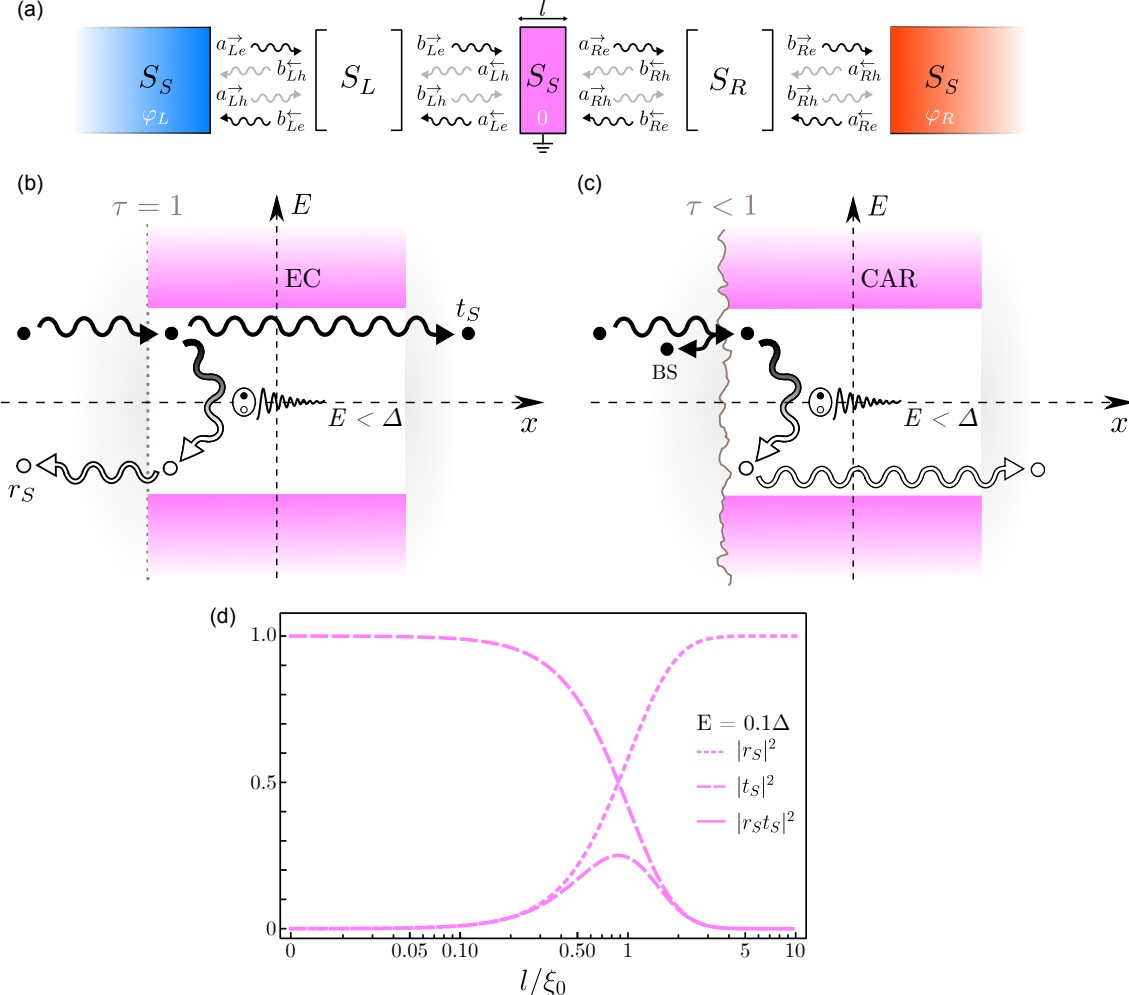

Figure 1: Scattering description of multi-channel Andreev molecules. (a) Plane waves corresponding to electrons (black arrows) and holes (gray arrows) scatter on superconductors (blue, red and magenta) and weak links. The left (right) superconductor has phase $\varphi_{L,R}$ and the central superconductor of length $l$ is grounded with phase zero. The ground connection allows applying the phase differences $\varphi_{L,R}$ independently by flowing different currents through each junction. The matrix $S_S$ describes Andreev scattering processes on the superconductors while $S_{L,R}$ describes normal scattering at the weak links. Only one channel is sketched. (b) For $l \lesssim \xi_0$, an electron incident on the central superconductor with energy $E$ less than the superconducting gap $\Delta$ can transmit across (elastic co-tunneling, EC) with probability amplitude $t_S$ or be Andreev reflected as a hole with amplitude $r_S$. (c) In the presence of scattering, such that the left channel's transmission probability is $\tau < 1$, the electron may also be backscattered (BS) or undergo crossed Andreev reflection (CAR), which converts it into an outgoing hole. (d) The probabilities of Andreev reflection, transmission, and their product, which factors into the probability for CAR, is plotted as a function of $l/\xi_0$ for fixed energy $E = 0.1\Delta$. In a long superconductor, $l \gg \xi_0$, only Andreev reflection occurs whereas for a short one, $l \ll \xi_0$, only elastic co-tunneling occurs. At intermediate values $l \approx \xi_0$ and in the presence of scattering there is a peak in the CAR probability which goes to zero elsewhere.

The scattering equation for the weak links is given by $\mathcal{B} = S_N \mathcal{A}$ with

$$S_N = \begin{pmatrix} S_L & 0 & 0 & 0 \\ 0 & S_R & 0 & 0 \\ 0 & 0 & S_L^* & 0 \\ 0 & 0 & 0 & S_R^* \end{pmatrix}. \qquad (2)$$

The individual normal scattering matrices at the left and right weak links are $S_{L,R}$ for electrons and $S_{L,R}^*$ for holes. The specific form of $S_{L,R}$ and $S_{L,R}^*$ will depend on the weak links. For example the scattering matrix corresponding to a Dirac $\delta$-potential as used in the BdG analysis of the Andreev molecule [5] is given by

$$S = \begin{pmatrix} \frac{-iu}{1+iu} & \frac{1}{1+iu} \\ \frac{1}{1+iu} & \frac{-iu}{1+iu} \end{pmatrix}, \qquad (3)$$

where the constant $u$ is related to the strength of the $\delta$-potential, $U_0$, and the Fermi velocity, $v_F$, by $u = U_0/\hbar v_F$. For simplicity, in the following analysis for a multi-channel weak link we use random symmetric unitary matrices for $S_L$ and $S_R$. These matrices can in principle include additional scattering at the superconductor-weak link interface. Other classes of scattering matrices corresponding to breaking time-reversal symmetry or spin-rotation symmetry can be used to model the effect of a magnetic field or spin-orbit interaction [12–14]. The dimensions of $S_N$ is $8N \times 8N$ where $N$ is the number of channels.

It remains to determine scattering on the superconductors. In contrast to scattering at the normal weak links, which need not preserve momentum, scattering on the superconductors occur through Andreev processes which are momentum-conserving when the Fermi energy is much larger than the superconducting gap.

For the semi-infinite superconducting electrodes to the left and right, for energies smaller than the superconducting gap ($|E| < \Delta$), the only scattering process possible is Andreev reflection, in which an incident electron is retroreflected as a hole and an incident hole is retroreflected as an electron. This Andreev reflection amplitude is $r_A = e^{-i(\alpha \pm \varphi_{L,R})}$, where $\varphi_{L,R}$ is the superconducting phase of the left (right) superconductor and $\alpha = \cos^{-1}\epsilon$ with $\epsilon = E/\Delta$ [15]. Since the Andreev reflection probability, $|r_A|^2$, is unity the semi-infinite electrodes act as perfect phase-conjugating mirrors for electrons and holes [16]. The phase shift acquired in reflection is the sum of $\alpha$, which is energy dependent, and the superconducting phases $\varphi_{L,R}$.

As shown in Fig. 1(b), the situation is different for a superconductor of finite length, in which an electron or hole can also propagate across and emerge on the other side without being retroreflected. For example in Fig. 1(b) an electron incident on the central superconductor from the left with amplitude $b_{Le}^{\rightarrow}$ and momentum $+k_F$ may either be retroreflected as a left propagating hole of amplitude $a_{Lh}^{\leftarrow}$ or transmitted as a right propagating electron of amplitude $a_{Re}^{\rightarrow}$, both particles having momentum $+k_F$.

When there is normal scattering in addition to a finite superconductor, such as in Fig. 1(c) where the weak link has transmission probability $\tau < 1$, electrons and holes can also be backscattered (BS) and crossed-Andreev reflected (CAR), which consists of tunneling through the superconductor and conversion from electron to hole or vice-versa [17]. As depicted the CAR process for an electron incident from the left corresponds to first an Andreev reflection and then backscattering of the retroreflected hole, which then traverses the finite superconductor and exits toward the right. This mechanism can also be seen as the formation, in the central slab, of a Cooper pair comprised of electrons from both left and right electrodes. The time-reversed equivalent is known as Cooper-pair splitting. The CAR process, which does not conserve momentum, requires backscattering in the normal weak links.

The probability amplitude associated with the process of partial Andreev reflection, Fig. 1(d), can be found using the continuity of wavefunctions at each interface. These wavefunctions are built from the electron and hole eigenstates of an infinite superconductor ($\eta = e$

or $h$),

$$\psi^{\varphi}_{\eta\pm}(x) = \left(u^{\varphi}_{\eta}, v^{\varphi}_{\eta}\right)^{T} e^{\pm i k_{\eta} x}, \tag{4}$$

where the coherence factors are given by

$$u^{\varphi}_{e,h} = \frac{e^{-i\varphi/2}}{\sqrt{2}} \left(1 \pm \sqrt{1-\epsilon^{-2}}\right)^{1/2},$$

$$v^{\varphi}_{e,h} = \text{sgn}(\epsilon) \frac{e^{i\varphi/2}}{\sqrt{2}} \left(1 \mp \sqrt{1-\epsilon^{-2}}\right)^{1/2},$$

and $k_{e,h}$ are complex to account for bound states. If the superconducting gap is much smaller than the Fermi energy $\Delta \ll E_F$, they can be approximated as $k_{e,h} \approx k_F \pm i/\xi$ where $k_F$ is the Fermi momentum in the normal state and the coherence length is a function of energy $\xi^{-1} = \xi_0^{-1}\sqrt{1-\epsilon^2} \ll k_F$. Here $\xi_0 = \hbar v_F/\Delta$ is the bare superconducting coherence length, $v_F$ is the Fermi velocity and $\epsilon = E/\Delta$ is the normalized energy.

If we focus on the subspace of waves with positive momentum the wavefunction is given by

$$\psi(x) = \begin{cases} b^{\rightarrow}_{Le} e^{ik_F\left(x+\frac{l}{2}\right)}(1,0)^{T} + a^{\leftarrow}_{Lh} e^{ik_F\left(x+\frac{l}{2}\right)}(0,1)^{T}, & x < -l/2, \\ c^{+}_{e} e^{ik_e x}(u^0_e, v^0_e)^{T} + c^{+}_{h} e^{ik_h x}(u^0_h, v^0_h)^{T}, & |x| \le l/2, \\ a^{\rightarrow}_{Re} e^{ik_F\left(x-\frac{l}{2}\right)}(1,0)^{T} + b^{\leftarrow}_{Rh} e^{ik_F\left(x-\frac{l}{2}\right)}(0,1)^{T}, & x > l/2, \end{cases}$$

where the three regions are the finite superconducting slab ($|x| \le l/2$) and the normal conductors to the left ($x < -l/2$) and right ($x > l/2$) of the slab. The superconducting phase on the central superconductor is fixed at zero and serves as the reference for the phase differences $\varphi_{L,R}$ on the left and right superconductors. Each junction can be shorted by a superconducting loop which allows tuning $\varphi_{L,R}$ independently with external magnetic fields. In addition this ground connection allows an additional path for current flow such that the supercurrents through the two weak links may be different.

In the normal regions ($x < -l/2$ or $x > l/2$) only electron or hole plane waves are possible, with wavevectors $\pm k_F$ and coherence factors either $(1,0)$ (electrons) or $(0,1)$ (holes). In the superconducting slab the wavefunctions mix electrons and holes and may have an exponential, energy-dependent envelope as a result of the complex wavevectors $k_{e,h}$.

Imposing boundary conditions at the slab edges $x = \pm l/2$ to preserve continuity we have

$$\begin{pmatrix} b^{\rightarrow}_{Le} \\ a^{\leftarrow}_{Lh} \end{pmatrix} = e^{-\frac{ik_F l}{2}} \begin{pmatrix} u^0_e e^{l/2\xi} & u^0_h e^{-l/2\xi} \\ v^0_e e^{l/2\xi} & v^0_h e^{-l/2\xi} \end{pmatrix} \begin{pmatrix} c^{+}_{e} \\ c^{+}_{h} \end{pmatrix},$$

$$\begin{pmatrix} a^{\rightarrow}_{Re} \\ b^{\leftarrow}_{Rh} \end{pmatrix} = e^{+\frac{ik_F l}{2}} \begin{pmatrix} u^0_e e^{-l/2\xi} & u^0_h e^{l/2\xi} \\ v^0_e e^{-l/2\xi} & v^0_h e^{l/2\xi} \end{pmatrix} \begin{pmatrix} c^{+}_{e} \\ c^{+}_{h} \end{pmatrix}.$$

By eliminating the coefficients $c^{+}_{e,h}$ we can relate incoming and outgoing waves with a scattering matrix,

$$\begin{pmatrix} a^{\leftarrow}_{Lh} \\ a^{\rightarrow}_{Re} \end{pmatrix} = \begin{pmatrix} r_S & t^{-}_{S} \\ t^{+}_{S} & r_S \end{pmatrix} \begin{pmatrix} b^{\rightarrow}_{Le} \\ b^{\leftarrow}_{Rh} \end{pmatrix},$$

where we define the Andreev transmission amplitude,

$$t_S = \frac{e^{-l/\xi}\left(1 - e^{-2i\alpha}\right)}{1 - e^{-2l/\xi}e^{-2i\alpha}}, \tag{5}$$

with $t^{\pm}_{S} = t_S e^{\pm ik_F l}$, and the partial Andreev reflection amplitude,

$$r_S = \frac{e^{-i\alpha}\left(1 - e^{-2l/\xi}\right)}{1 - e^{-2l/\xi}e^{-2i\alpha}}. \tag{6}$$

For the negative momentum wavefunction the substitution $k_F \rightarrow -k_F$ yields the same scattering matrix with $t_S^+$ and $t_S^-$ swapped.

These amplitude satisfy $|r_S|^2 + |t_S^{\pm}|^2 = 1$ as expected from quasiparticle conservation. In a realistic system with a three-dimensional central superconductor, the wavefunctions $\psi$ (Eq. (4)) will be spherical, the longitudinal part of the wavevector can take any value between 0 and $k_F$, and the geometric factors $e^{-l/\xi}$ describing the envelope of the probability amplitudes $t_S, r_S$ (Eqs. (5) and (6)) will be different. In general the envelope will decay faster and acquire additional dependence on the Fermi wavelength or the mean free path [18–20]. This reduction can be understood from the increase in scattering angle as the number of dimensions is increased.

The following analysis is limited to the one-dimensional case. For convenience and visibility we set $k_F l$ to constant values in the scattering coefficients while maintaining $k_F \gg 1$. In principle each channel may have a different phase factor resulting from interference but such offsets are already included via the random unitary scattering matrices $S_{L,R}$ and do not change the results qualitatively. In addition we have assumed that the energy gap of the superconducting slab is the same as that of the superconducting electrodes, effectively ignoring any inverse proximity effect which is reasonable given that we consider typical semiconducting weak links.

In Fig. 1(d) we plot the Andreev reflection probability $|r_S|^2$ and transmission probability $|t_S|^2$ for fixed energy $\epsilon = 0.1$ as a function of $l/\xi_0$. The likelihood of elastic co-tunneling (EC), Fig. 1(b), in the absence of scattering at the weak links ($\tau = 1$) is quantified by $|t_S|^2$. As the superconductor thickness goes to zero, $l/\xi_0 \rightarrow 0$, Andreev reflections are suppressed and all quasiparticles tunnel across, $t_S \rightarrow 1$. Andreev processes are equally probable when $l/\xi_0 \approx 1$. As we extend the length of the central superconductor, $l/\xi_0 \rightarrow \infty$, one recovers the Andreev reflection amplitude of a semi-infinite superconductor, $r_S \rightarrow r_A = e^{-i\alpha}$, and transmission is quashed, $t_S \rightarrow 0$. The Andreev phase-conjugating mirror is only perfect if it is much thicker than $\xi_0$, the characteristic length scale for Andreev reflection.

Scattering at the weak links will also reduce elastic co-tunneling. If the single-channel transmissions of the weak links are $\tau_{L,R}$, the first order EC probability will be reduced to $\tau_L \tau_R |t_S|^2$. For $\tau < 1$, there will be higher order processes involving multiple reflections at the barriers which will also transmit a particle across the superconductor.

Also plotted in Fig. 1(d) is the probability $|r_S t_S|^2$, which is the Andreev scattering contribution to the first-order CAR process depicted in Fig. 1(c). If the left interface has transmission probability $\tau < 1$, this CAR process requires normal barrier transmission ($\tau$), an Andreev reflection ($|r_S|^2$), a normal reflection ($1-\tau$), and an Andreev transmission ($|t_S|^2$). The Andreev contribution, $|t_S r_S|^2$, is maximal at 0.25 for a separation $l/\xi_0$ such that $|t_S| = |r_S| = 0.5$ and the maximum of the normal part, $\tau(1-\tau)$, is also 0.25 for $\tau = 0.5$. Therefore the maximum likelihood of the first-order CAR process is 6.25%, with higher order processes contributing little as they scale as $\tau^n(1-\tau)^n$. Ignoring higher order processes the likelihood of EC in the presence of scattering at the left weak link, $\tau|t_S|^2$, is approximately four times that of CAR for $\tau = 0.5$ and at a comparable separation $l/\xi_0 \lesssim 1$ such that $|t_S|^2 \approx 0.5$. The optimal separation $l/\xi_0$ to maximize CAR and EC depends on the energy $\epsilon$ but the relative likelihood for CAR over EC remains $(1-\tau)/2$. In a symmetric situation where both weak links have transmission $\tau$, the first-order expressions above are reduced by a factor $\tau$.

In a similar fashion to the derivation of $S_N$, we use these results for scattering from the three superconductors to define a matrix $S_S$ which relates waves incident on the slab ($\mathcal{B}$) to the outgoing waves, $\mathcal{A} = S_S \mathcal{B}$,

$$S_S = \begin{pmatrix} S_{ee} & S_{eh} e^{-i\Phi} \\ S_{eh} e^{i\Phi} & S_{hh} \end{pmatrix} \otimes \mathbb{I}_N,$$

with blocks $S_{eh}$ on the anti-diagonal for Andreev reflections,

$$S_{eh} = \begin{pmatrix} r_A & 0 & 0 & 0 \\ 0 & r_S & 0 & 0 \\ 0 & 0 & r_S & 0 \\ 0 & 0 & 0 & r_A \end{pmatrix},$$

and blocks $S_{ee}$ and $S_{hh}$ on the diagonal for tunneling through the central superconducting slab,

$$S_{ee} = \begin{pmatrix} 0 & 0 & 0 & 0 \\ 0 & 0 & t_S^+ & 0 \\ 0 & t_S^+ & 0 & 0 \\ 0 & 0 & 0 & 0 \end{pmatrix}.$$

$S_{hh}$ is obtained from $S_{ee}$ with the transformation $t_S^+ \to t_S^-$. The superconducting phases are contained in the diagonal matrix $\Phi = \text{diag}(\varphi_L, 0, 0, \varphi_R)$ and $\mathbb{I}_N$ is the $N \times N$ identity matrix. The total size of $S_S$, like $S_N$, is $8N \times 8N$, accounting for $N$ conduction channels.

We combine the scattering equation for weak links, $\mathcal{B} = S_N \mathcal{A}$, and for superconductors, $\mathcal{A} = S_S \mathcal{B}$, in order to obtain the master equation,

$$\mathcal{B} = S_N S_S \mathcal{B}. \tag{7}$$

The scattering product $S_N S_S$ depends on energy $\epsilon$, the scattering properties of the weak links ($S_{L,R}$), and the superconducting phases $\varphi_{L,R}$. Eq. (7) is a unity eigenvalue problem in which solutions of the characteristic equation,

$$\det\left(\mathbb{I}_{8N} - S_N S_S\right) = 0, \tag{8}$$

gives the energy spectrum $\epsilon$, the scattering amplitudes $a$ and $b$, and the corresponding wavefunctions of the Andreev molecule [21].

To verify correctness we numerically solved Eq. (8) for the spectra in the case of a single channel Andreev molecule with symmetric $\delta$-function barriers, i.e. $S_L$ and $S_R$ given by Eq. (3), and compared for agreement with the Bogolubiov-de-Gennes solution for the same parameters [5].

## 3 Energy Spectra

In Fig. 2 we show the evolution in the energy spectra of a multi-channel Andreev molecule as the size of the central superconductor is reduced. Spectra are obtained by numerically solving the characteristic equation Eq. (8) for fixed 20-channel random scattering matrices $S_{L,R}$ and fixed phase $\varphi_R = 3\pi/5$. Each channel of each weak link will have an effective transmission $\tau$ which can be extracted from the scattering matrices $S_{L,R}$. The spectra are plotted as a function of the left phase $\varphi_L$ for four values of the separation $l/\xi_0$. Each conduction channel of each junction hosts one pair of ABS and as a consequence there are $4N = 80$ lines, some of which are close to the gap edge and difficult to distinguish.

For large separation, $l/\xi_0 \gg 1$, there is no coupling between the two weak links, and the spectral lines follow the standard ABS energy dispersion,

$$E_{Ln,Rn}^\pm = \pm\Delta\sqrt{1 - \tau_{Ln,Rn}\sin^2\left(\varphi_{L,R}/2\right)},$$

where $\tau_{Ln,Rn}$ corresponds to the transmission of the $n$-th channel in the left or right weak link. Since the right phase is fixed, $\varphi_R = 3\pi/5$, ABS corresponding to the right weak link

(red) do not disperse with $\varphi_L$, whereas those of the left junction (blue) dip towards zero as $\varphi_L$ approaches $\pi$. There is no hybridization between ABS at the right and left junctions and the spectral lines cross without forming gaps.

As the junctions are brought closer, for $l/\xi_0 = 1, 0.5, 0.1$, multiple avoided crossings materialize, signaling the formation of Andreev molecules. Similarly to the one-channel case [5], the amplitude of the avoided crossings increases as the separation is reduced and some discrete states are gradually pushed out into the continuum.

At separation $l = 0.1\xi_0$, where the Andreev molecule fuses into a single weak link, only approximately half of the ABS remain in the gap and the states have shifted in phase to the right by $\varphi_R = 3\pi/5$.

Overall the spectra of Fig. 2 for the multi-channel case show qualitatively the same behavior as for the Andreev molecule in the single channel case [5]. The most obvious global sign of hybridization remains the breaking of symmetry about the phase $\varphi_L = \pi$. Since there are often phase offsets in experiments it is difficult to verify that $\varphi_L = \pi$. One could instead check for symmetry about the more easily identifiable point, $\varphi_L = \varphi_L^0$, where the ABS are closest to zero in energy at a fixed phase $\varphi_R$. The multi-channel spectra indicate that the symmetry point $\varphi_L^0$ shifts from $\pi$ to $\pi + \varphi_R$ as the separation $l/\xi_0$ goes from infinity to zero and that symmetry is broken for $l \lesssim \xi_0$. Even though the spectra will become more dense as the number of channels is increased, this symmetry breaking will be relevant experimentally as long as $l \lesssim \xi_0$.

## 4 Molecular Bound States

An eigenvector $\mathcal{B}_0$ which solves Eq. (7) corresponds to a closed trajectory or bound state of the Andreev molecule, formed due to interference between propagating and counterpropagating waves. There are three different types of closed cycles, or orbits, with two non-trivial ones which can be built from the EC and CAR processes shown in Fig. 1.

The trivial cycle is a conventional Andreev bound state at one of the weak links and is represented in Fig. 3(a) where the central superconductor is large, $l \gg \xi_0$. The closed orbit consists of two Andreev reflections at the right weak link, with the left moving hole of amplitude $b_{Rh}^{\leftarrow}$ being completely transformed into a right moving electron of amplitude $b_{Re}^{\rightarrow}$ at the central superconductor (purple). Since the Andreev transmission probability $t_S$ vanishes for large $l/\xi_0$, Fig. 1(d), the incident hole cannot be transmitted through the central superconductor. Likewise at the infinite left (blue) and right (red) superconductors, only Andreev reflection is possible. A conventional ABS does not connect particles on all three superconductors and therefore the supercurrent associated with it only flows between two superconductors.

With a shorter central superconductor, Fig. 3(b), one has the first non-trivial or "molecular" Andreev bound state: the loop passing through all three superconductors. This orbit consists of two simultaneous EC processes, one shown in Fig. 1(b), and the other its particle-conjugate dual in which a hole propagates from right to left. Such a double elastic cotunneling (dEC) process transports two electrons from the left to right superconductor. Since the phases are fixed and all voltages are zero, this charge transfer corresponds to a unidirectional supercurrent flowing across the device. dEC-type bound states are probable when the normal scattering matrices have high channel transmissions and the phases $\varphi_{L,R}$ have opposing signs and values which result in an energy degeneracy in the limit $l/\xi_0 \to \infty$. In the case of a symmetric single-channel Andreev molecule [5], dEC is maximal when the phases satisfy $\varphi_L = -\varphi_R$.

Fig. 3(c) shows the dEC bound state probability as a function of $l/\xi_0$ determined by numerically solving the eigenvalue problem, Eq. (7), for the lowest positive energy state of a symmetric, single-channel Andreev molecule of transmission $\tau \approx 0.94$. In red we plot the probabilities $|b_{Re}^{\rightarrow}|^2$ and $|b_{Rh}^{\leftarrow}|^2$ corresponding to the orbit shown in Fig. 3(a) or the right part

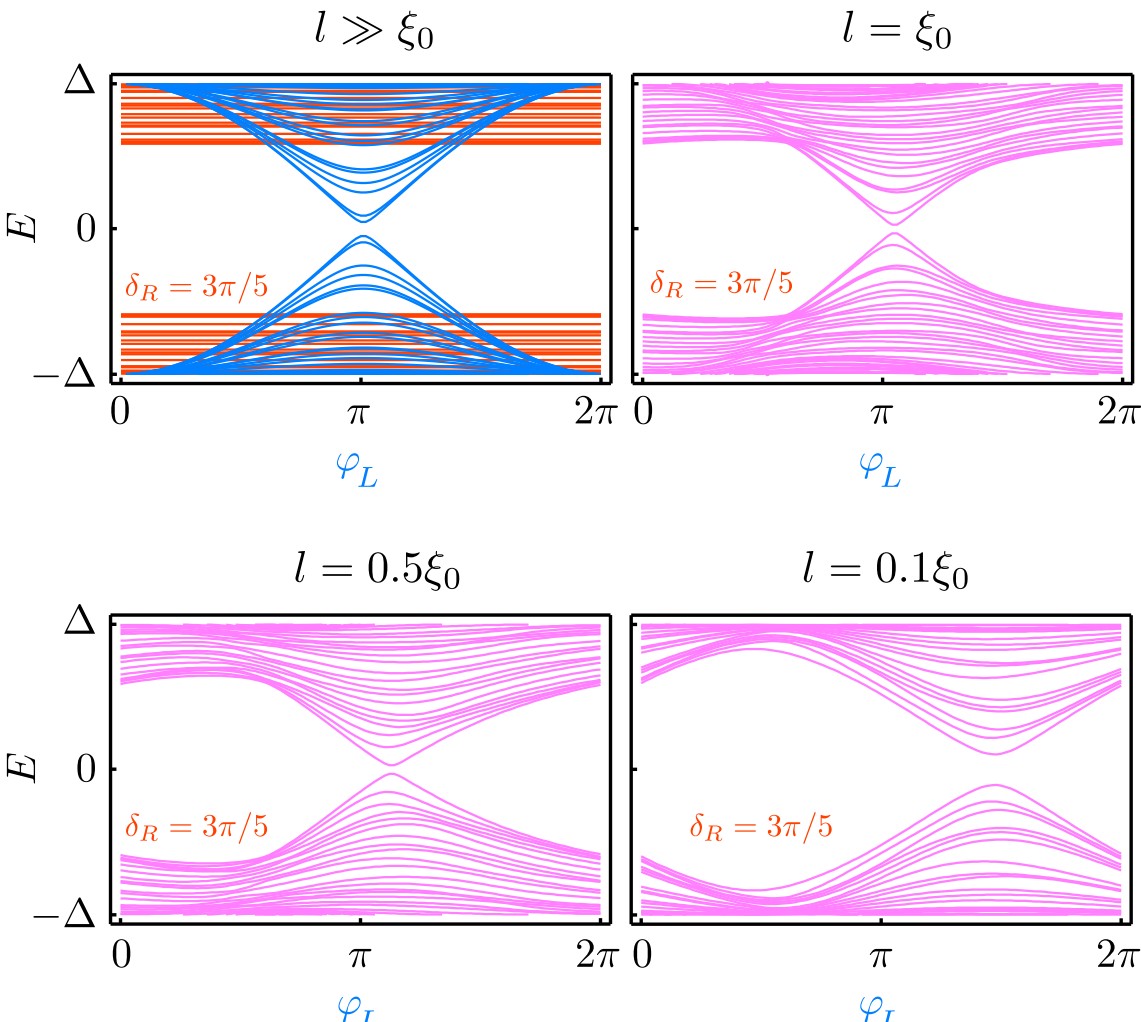

Figure 2: Energy spectra of a multi-channel Andreev molecules as a function of separation $l/\xi_0$. Scattering for twenty-channel left and right weak links is described by randomly generated symmetric unitary matrices $S_L$ and $S_R$ which are the same for each value of $l/\xi_0$. The superconducting phase on the right weak link is fixed at $\varphi_R = 3\pi/5$ and the left phase, $\varphi_L$, is varied. For $l \gg \xi_0$, the red lines in the spectrum corresponds to Andreev Bound States (ABS) localized at the right weak link and independent of $\varphi_L$, whereas the blue lines correspond to ABS localized on the left weak link. The spectral lines are distinct because the effective transmission of each channel, determined by $S_{L,R}$, is random. As the separation $l/\xi_0$ is reduced, the red and blue lines, now purple, form avoided crossings indicating the hybridization of Andreev states and the formation of an Andreev molecule. For small separation $l \ll \xi_0$ the spectrum transforms into that of a single twenty-channel weak link shifted by $\varphi_R$. Note that $S_L \neq S_R$ and for convenience the momentum is chosen such that $k_F l = 0 \pmod{2\pi}$, with $k_F l \gg 1$.

of Fig. 3(b). In blue we plot $|b_{Le}^{\rightarrow}|^2$ and $|b_{Lh}^{\leftarrow}|^2$ which corresponds to the complementary orbit passing through the left weak link in Fig. 3(b). The eigenvectors are normalized so that the probabilities sum to 1 and the amplitudes $a$ are related to the $b$'s by the scattering matrix $S_N$. To maximize dEC, the phases are fixed at $\varphi_R = 0.5\pi$ and $\varphi_L = -0.48\pi$. The slight detuning of $\varphi_L$ from $-0.5\pi$ allows being sufficiently far from degeneracy such that there is no mixing between left and right eigenstates at $l/\xi_0 = 10$. In principle at exact degeneracy and arbitrarily large $l/\xi_0$ a viable eigenstate can consist of equal weights at the left and right weak links.

At large separation, $l/\xi_0 \approx 10$, both probabilities at the right weak link (red) are approximately 0.5 whereas those at the red weak link (blue) are almost zero, indicating that the eigenstate is a conventional ABS as in Fig. 3(a).

As the separation is reduced, the weights at the left weak link (blue) start to increase and those at the left weak link (red) decrease, indicating the formation of a dEC state. The position of the step will depend on the detuning of $\varphi_L$ from $-\varphi_R$. Near $l/\xi_0 \approx 1$, the orbit is approximately equally distributed between the left and right weak links. The decomposition of dEC into two simultaneous EC processes leads to the qualitatively similar form of the probabilities in blue with the EC probability $|t_S|^2$ of Fig. 1(d).

For even smaller separation both the red and blue probabilities decrease and are compensated by an increase in the amplitudes $|b_{Le,Re}^{\leftarrow}|^2$ and $|b_{Lh,Rh}^{\rightarrow}|^2$ (not shown) of the counter-propagating orbit given by reversing the directions of the arrows in Fig. 3(b). The relative weight of these two trajectories will be determined by the value of the phase difference $\varphi_R$. This can be understood by considering the complementary time-reversed ABS trajectory to the one shown in Fig. 3(a). When the phase $\varphi_R$ is zero or $\pi$, such that the supercurrent is zero, these two trajectories have equal weights and compensate each other. At extrema of the supercurrent one trajectory will dominate. This is why with our choice of $\varphi_R = \pi/2$ the red probabilities in Fig. 3(c) approach 0.5 for large $l/\xi_0$, near a supercurrent maximum for the right weak link. The situation is similar for a dEC orbit and when the separation approaches zero, the total phase drop across the device is $\varphi_R - \varphi_L \approx \pi$, so the dEC supercurrent vanishes and both trajectories coexist. This is why all probabilities approach $1/8$ near $l/\xi_0 = 0$ in Fig. 3(c), resulting in approximately equal clockwise and counter-clockwise orbits. The additional splitting of the blue lines results from normal scattering and is absent when $\tau = 1$.

The second molecular bound state, dCAR, is shown in Fig. 3(d), and with respect to the dEC orbit involves two additional quasiparticle conversions in the central superconductor and a reversal of current direction at the left weak link. During the conversion an incident electron of energy $E$ is reflected as a hole of energy $-E$ which results in the crossing of trajectories at the central superconductor and the twist relative to the dEC diagram Fig. 3(b). dCAR describes supercurrent flowing from the central superconductor to the outer ones and cannot occur for a floating central island, or without a connection to ground.

The dCAR probability is plotted in Fig. 3(d) for the same $\varphi_R = \pi/2$ but with $\varphi_L = 0.52\pi \approx \varphi_R$ in order to maximize the effect while maintaining a detuning to avoid a trivial degeneracy. Note that although the probabilities in red are identical to those for dEC, Fig. 3(c), the probabilities in blue are $|b_{Le}^{\leftarrow}|^2$ and $|b_{Lh}^{\rightarrow}|^2$ to take into account the reversal of the trajectory on the left weak link. There is a non-physical numerical instability at exactly $l/\xi_0 = 0$ so the x-axis extends from $l/\xi_0 = 0.05$ to 10. As expected at large separation $l/\xi_0 = 10$ the eigenstate is an ABS localized at the right weak link.

As the separation is reduced the probability shifts to the left weak link, much as with dEC. The increase in probability at the left weak link (blue lines) occurs at smaller $l/\xi_0$ than for dEC, most likely a result of the high value of transmission which leads to weak dCAR hybridization. After reaching a maximum at $l/\xi_0 \approx 1$ the blue lines take a sharp downturn and approach zero as the separation is further reduced. The probability for dCAR follows the Andreev reflection probability which vanishes as $l/\xi_0 \to 0$. As with dEC the probabilities describing propagation

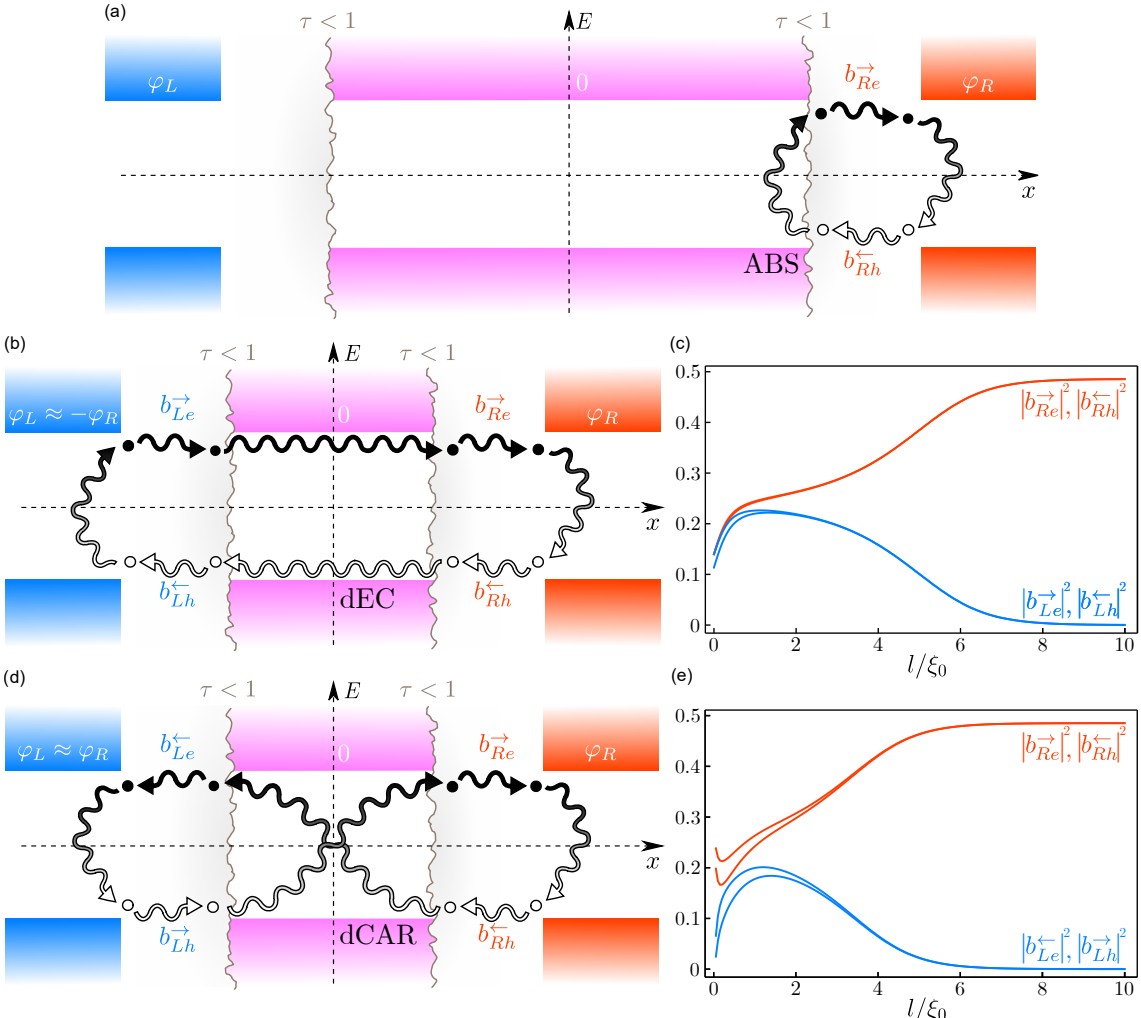

Figure 3: Bound states of an Andreev molecule. (a) At large separation $l \gg \xi_0$ the only eigenstate is a conventional Andreev Bound State (ABS), shown here localized at the right weak link by Andreev reflections at the central (purple) and right (red) superconductor. (b) At small separation $l \lesssim \xi_0$ and for superconducting phases $\varphi_L \approx -\varphi_R$ there is an additional trajectory, double Elastic Co-tunneling (dEC), which extends across all three superconductors. (c) The likelihood of dEC (blue lines) and ABS (red lines) trajectories are plotted as a function of separation $l/\xi_0$ for $\varphi_R = 0.5\pi$, $\varphi_L = -0.48\pi$, $\tau \approx 0.94$ and $k_F l \gg 1, k_F l = 0 \pmod{2\pi}$. The dEC probability increases as the separation is reduced. (d) A second "molecular" trajectory extending across all superconductors is possible at small separation $l \lesssim \xi_0$ but for superconducting phases $\varphi_L \approx \varphi_R$. This is called double Crossed Andreev Reflection (dCAR) and differs from dEC by additional Andreev reflections in the central superconductor. (e) The likelihood of dCAR and ABS trajectories are plotted as a function of $l/\xi_0$. Parameters are the same except for $\varphi_L = 0.52\pi$ and $k_F l = \pi/2 \pmod{2\pi}$. The dCAR probability vanishes for large and small separation and is maximal at $l \approx \xi_0$. Results are obtained by numerically solving the eigenvalue equation, Eq. (7).

through the right weak link, including the time-reversed ones not shown, approach approximately the same value as $l/\xi_0 \to 0$. However since the probability of all trajectories at the left weak link must vanish, the red lines approach a value of $1/4$ instead of $1/8$ as with dEC. The additional splitting of the probabilities for $l/\xi_0 \lesssim 1$ is also due to imperfect transmission. Unsurprisingly, the overall shape of the dCAR probabilities (blue lines) are similar to that of the CAR probability plotted in Fig. 1(d).

In the general multi-channel, non-symmetric case and as a function of the separation the eigenstates will be mixtures of conventional ABS and molecular ABS. The phase configuration necessary for molecular orbits will coincide with the position of level crossings in the large separation ABS energy spectrum such as in Fig. 2 for $l = 10\xi_0$.

# 5 Conclusion

Andreev molecules, or in general, arbitrary mesoscopic systems with superconducting regions of size comparable to the coherence length can be effectively modeled with the scattering approach incorporating the partial Andreev reflection and transmission coefficients ($r_S, t_S$). We validated this formalism by checking for agreement with the Bogolubiov-de-Gennes results for a single-channel Andreev molecule [5]. We then calculated the energy spectrum of a multi-channel Andreev molecule, modeling the experimentally relevant system of an epitaxial superconductor/semiconductor nanowire with nanoscale weak links. The calculations show that Andreev Bound State hybridization is robust and leads to observable consequences even in multi-channel mesoscopic systems. In addition we have shown how to interpret the formation of Andreev molecules in terms of the microscopic non-local scattering processes of double elastic co-tunneling and double crossed Andreev reflection. We quantified the probability for these processes and determined the conditions to maximize them.

Although the formalism presented here has the advantage of simplicity, it has several limitations. Our one-dimensional treatment ignores the lateral extension of the central superconductor which, as mentioned above, results in a larger overlap between ABS than expected in three dimensions. A smaller overlap will lead to a reduction in the size of the avoided crossings in Fig. 2 as well as reducing the probabilities for dEC or dCAR states in Fig. 3. However an analysis for a 3D finite superconductor has shown that the energy gaps due to hybridization will remain measurably large, if not a significant fraction of $\Delta$ [20]. We have also confined our treatment to short weak links in which there is no additional accumulated phase. A sophisticated treatment incorporating the quality of the nanowire-superconductor contact as well as the lead resistance has attacked some of these shortcomings and elucidated in detail the impact of the central lead on ABS hybridization [22].

The scattering formalism can easily be extended to more complicated structures and take into account additional mechanisms such as spin-orbit interactions or a magnetic field, relevant for Majorana bound states. It would also be possible to model superconducting weak links with multi-junction nanowires [23], where a quasiparticle incident on a short superconductor could be Andreev transmitted in multiple directions. Yet another topology is Andreev polymers, systems with chains or networks of short superconducting segments connected by weak links, which would allow ABS hybridization across several sites.

# Acknowledgements

We acknowledge insightful discussions with Yuli Nazarov and support from Jeunes Equipes de l'Institut de Physique du Collège de France.

**Funding information**   This research was supported by IDEX grant ANR-10-IDEX-0001-02 PSL and a Paris "Programme Emergence(s)" Grant. This project has received funding from the European Research Council (ERC) under the European Union's Horizon 2020 research and innovation programme (grant agreement 636744).

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
