# Peer review of "Scattering description of Andreev molecules"

_SciPost Physics, doi:SciPost Phys. Core 2, 009 (2020)_

## Round 1 · Referee Report · Anonymous (Referee 1) · 2020-3-10

Report
The authors consider a superconducting wire interrupted by two weak links. They study the spectrum of the central region, which they refer to as an "Andreev molecule". The manuscript builds on an earlier paper by the same group of authors (arXiv:1809.11011) in which they studied a single-mode wire, while here they consider the multi-mode case. The scattering description which they employ is routine in the field and the phenomenon of crossed Andreev reflection which is obtained is also very well studied and understood. This limits the novelty and significance of the work, but should not by itself prevent publication.
I have, however, also several concerns regarding the scientific validity, which do stand in the way of publication.
-
The central region of the superconducting wire (the "Andreev molecule") is grounded, and the authors say that this allows them to fix the superconducting phase of the central region at 0. Here they are confusing voltage bias and flux bias. "Grounding" means that the voltage is zero, it does not mean that the phase is zero. The phase difference phi_L-phi_R between the outer ends of the wire can be fixed by a flux bias, but the phase of the central region should then be determined selfconsistently, it is not fixed by the voltage ground.
-
The authors assume that the electron and hole propagate through the central region with longitudinal momentum k_e,h = k_F ± i/ξ. In a multi-mode junction the longitudinal momentum can vary between 0 and the maximal value of k_F. Setting it equal to k_F for all modes does not seem justified.
-
On page 6 the authors write "we ignore fast phase oscillations in t_S and r_S arising from the small Fermi wavelength by fixing k_F l arbitrarily and independent of l". This assumption makes no sense to me. In a phase coherent treatment these phase oscillations should play a crucial role.
-
The plots are calculated by "randomly generated symmetric unitary matrices S_L and S_R". These matrices should represent the weak link, which I presume is a tunnel junction. Since no disorder is included in the superconductor, the randomness needed for this assumption is not present and I do not understand the justification for this choice.

Author: Çağlar Girit on 2020-03-19 [id 771]
Thank you for reading our manuscript and providing feedback.
It is correct that the scattering description we employ is widely used to model Josephson junctions and does not represent a technical challenge. Obtaining expressions of our scattering matrices is relatively straightforward but it is not the main message of our manuscript. The novelty of our work lies in reporting what a scattering description has to tell us about Andreev molecules:
a) Having multiple channels (Fig. 2) does not change qualitatively the spectrum of hybridized Andreev Bound States (ABS) compared to what was reported in our previous work for a single channel (arXiv:1809.11011),
b) The scattering amplitudes (Fig. 3) obtained for configuration of maximal hybridization, phi_L = - phi_R and phi_L = phi_R, demonstrates that the formation of bound states is almost exclusively dominated by a specific microscopic mechanism, dEC and dCAR respectively.
These conclusions are important for the experimental realization of Andreev molecules since it provides the appropriate conditions for observation of novel phenomena such as avoided crossings between ABS in the energy spectrum and non-local Josephson Effect. On top of that, it gives more intuition about the nature of the electronic states in an Andreev molecule.
We also agree that the Crossed Andreev Reflection (CAR) phenomenon has been extensively studied (we give [17] as a reference since it is one of the pioneering works but we are happy to add more references if necessary). We simply mention CAR in the introduction and believe that our subsequent description in terms of successive individual microscopic scattering events is enlightening and complementary.
We also would like to mention that, though the result is simple to obtain, it is only recently that one can find explicit and complete expression of the scattering matrix for a superconductor of finite thickness (see for example an initial version of our previous work arxiv:1809.11011v1, or, in 2019, PRR 1 033212 and arxiv:1912.10307). Our work only tackles specific questions about Andreev molecule, but we believe it brings novel insights and tools to understand this physics.
We have carefully read your 4 concerns and we believe that none of them call into question the scientific validity of our work. However in order to improve the clarity of the manuscript we propose to make the following changes which address your comments:
In our work, the phase of the central superconductor is arbitrarily set to zero. By gauge invariance, this does not imply any loss of generality. What matters physically are the phase differences between superconductors. Since the central phase is chosen to be zero, these phase differences are given by phi_L and phi_R.
We believe the confusion could come from the phrasing in the caption of Fig. 1:
“The ground connection allows applying the phase differences phi_L,R independently”
What we meant here is that the current can be different in each junctions since there is a ground in the middle. As a consequence, phi_L and phi_R can be set independently. We propose changing this sentence to:
“The ground connection allows applying the phase differences phi_L,R independently by flowing different current through each junctions”
The confusion could also come from a similar sentence in the text page 5:
“Because of the ground connection, there are effectively two loops connecting the left and right superconductors which allow tuning phi_L,R independently with external magnetic fields”
This is indeed confusing since the ground is actually not necessary to form loops between superconductors. We propose modifying it with:
“Each junction can be shorted by a superconducting loop which allow tuning phi_L,R independently with external magnetic fields.”
We propose adding the following sentence to the caption of Fig. 2:
“Note that k_F might be different in each conduction channel.”
Indeed interference plays an important role in the construction of bound states. If one could gradually increase the length l of the central superconducting part while keeping k_F constant, this would lead to very fast oscillations for most quantities. For example, the width of avoided crossings in the spectra of ABS would oscillate to their full magnitude for the dCAR at phi_L = phi_R and partially for the dEC at phi_L = -phi_R. For the dCAR, when the avoided crossing fully closes, it means there is no hybridization, which leads to the cancellation of certain coefficients.
We have attached two plots (dEC and dCAR) with k_F = 10/xi_0 in our reply to show the effect of interference. In practice, it would be much bigger but the oscillations would be too fast to be visible on the plots.
In our manuscript, we made the deliberate choice to maintain k_F*l = constant while varying l in Fig. 3 in order to show how bringing two Josephson junctions close to each other, within a few superconducting coherence lengths, leads to the transition from localized ABS to hybridized ABS. This also shows how dEC and dCAR gradually take over local microscopic mechanisms. We believe this transition is more visible without the oscillations caused by interference, but that is purely a pedagogical choice.
In order to make this clearer, we propose changing the following sentence:
“In addition we ignore fast phase oscillations in t_S and r_S arising from the small Fermi wavelength by fixing k_Fl arbitrarily and independent of l while maintaining k_F*l>>1.”
into
“Moreover, in order to improve visibility, we remove fast oscillations of the scattering coefficients arising from interference by fixing k_F*l to constant values. This corresponds to plotting the k_F-independent envelope of the coefficients."
The reason we choose random S_L and S_R for quantum conductors with many conduction channels (Fig. 2) is not to include disorder but rather to keep a very generic approach with no assumption made on the nature of the conduction channel. For example, they have no reason to be the same than in the central superconductor.
The disorder of the superconducting region is not included in the matrix S_S. It could perfectly be taken into account in our approach by adding in series with S_S two normal scattering matrices S_S^L and S_S^R, on its left and on its right. They would describe scattering at the left and right interfaces of the superconductor and disorder in between. Nevertheless, it is possible to combine S_S^L and S_L (or S_S^R and S_R) into a single scattering matrix describing the left (resp. right) quantum conductor, which would be formally equivalent to what is done in our manuscript.
A better description of disorder could be done using a tight-binding approach as was done in the initial version of our previous work (arxiv:1809.11011v1) or using a semi-classical approach in more recent works by the Nazarov group. These results shows that there is no qualitative change in the formation of an Andreev molecule when one introduces disorder. Disorder mainly leads to a weakening of the ABS hybridization because of a reduction of the superconducting length xi (xi is replace by sqrt[xi*mfp] where mfp is the mean free path). However, it should remain measurable as long as the junction are separated by a distance comparable to this new xi.
Attachment:
Reply.pdf

---

## Round 1 · Referee Report · Francois Lefloch (Referee 2) · 2020-3-20

Report
In the context of quantum information, new quantum circuits including hybrid systems (here with superconducting and normal materials) and various geometries are widely investigated. In that sense, the results obtained by the authors, if not being ground-breaking’s, are of real interest for the community.
The paper is very well organized. In the limit of the assumptions made by the authors (see 1rst referee’s report) , the description of the model and the underlying physics phenomenon are very well described. This article a very pedagogical and clear.
Few comments that, in my opinion, limit the overall impact of this publication.
1 - The authors present results for a fixed number of channels (N = 20). It would have been interesting to discuss the effect of the number of channels in more details. This can be of real use for practical realizations with gated semiconducting nanowires.
2 - The origin of hybridization is due to (inverse) proximity effect in the central superconductor and explains the L/xi decay dependence. This inverse proximity effect is known to be strong when the transparency at the S/N interface is good (close to 1). But at the same time, the superconducting gap is locally reduced. On the contrary, when the transparency is smaller, the superconducting gap is restored but (inverse) proximity effect is less efficient.
It is not clear to me how the quality of the S/N interface will change the results but clearly the interface transparency is a real issue in experiments.
Note that the situation here seems different than in [22] as transport occurs necessarily though the central 3 D superconductor.

---

## Round 2 · Referee Report · Anonymous (Referee 1) · 2020-4-7

Report
The purpose of this manuscript is to go beyond the previous work of the authors on a single-mode nanowire (Ref. 5), to include the effects of multiple modes. The manuscript falls short in this objective, because it does not treat the effects of intermode scattering in a realistic way. The authors admit as much in their response, but claim that a more realistic treatment would not have made much difference. I disagree.
A key feature of a multimode nanowire is that tunnel probabilities can vary strongly from one mode to the other, depending on the longitudinal momentum. This effect is totally disregarded. Furthermore, a single-mode wire is strongly susceptible to localization, while in a multi-mode wire the localisation length can be much longer than the mean free path. This effect is also ignored. If this would be the first publication on this topic, then a "toy model" such as Ref. 5 would be acceptable, but this field has moved much beyond that stage. The analysis presented does not provide a reliable description of a multimode nanowire.
This is my main objection. A secondary point is the assumption of individually controllable phase differences which the authors assume. They write that "Each junction can be shorted by a superconducting loop which allow tuning phi_L,R independently with external magnetic fields." I do not understand how this might work, since they assume the central region is grounded, so if one would also short the junctions the entire wire would be grounded.
A key feature of a multimode nanowire is that tunnel probabilities can vary strongly from one mode to the other, depending on the longitudinal momentum. This effect is totally disregarded. Furthermore, a single-mode wire is strongly susceptible to localization, while in a multi-mode wire the localisation length can be much longer than the mean free path. This effect is also ignored. If this would be the first publication on this topic, then a "toy model" such as Ref. 5 would be acceptable, but this field has moved much beyond that stage. The analysis presented does not provide a reliable description of a multimode nanowire.
This is my main objection. A secondary point is the assumption of individually controllable phase differences which the authors assume. They write that "Each junction can be shorted by a superconducting loop which allow tuning phi_L,R independently with external magnetic fields." I do not understand how this might work, since they assume the central region is grounded, so if one would also short the junctions the entire wire would be grounded.

Author: Çağlar Girit on 2020-04-17 [id 796]
(in reply to Report 1 on 2020-04-07)Our approach actually treats the multi-mode scattering in the most general way. The scattering between modes depends on non-diagonal coefficients of matrices $S_L$ and $S_R$. We chose these matrices randomly from the Gaussian orthogonal ensemble (symmetric unitary matrices) representing a generic time-reversal invariant weak link. We do not specify the nature of these modes because all that is necessary for our analysis is that certain modes are highly transmitting ($\tau > 0.5$). Andreev molecules with modes of such high transmissions result in large avoided crossings ($> 0.05\Delta$) in the spectra. So even though we may miss interesting physical effects specific to each possible type of quantum conductors, our results are qualitatively the same as long as there are highly transmitted channels--which is the case for almost all experimentally relevant quantum conductors.
We did not say that.
It is true that each specific quantum conductor forming the weak links in the Josephson junction will induce differences in the spectra. However, for the features that are of interest for us, i.e. the formation of avoided crossings and an asymmetry in the spectrum with respect to $\pi$, are completely generic. These effects will simply be of different magnitude depending on the actual quantum conductor.
That is completely correct but this effect is not disregarded in our approach. The probability of scattering between modes can indeed vary strongly depending on the longitudinal momentum, the presence of disorder, the shape and cleanliness of interfaces between superconductors and weak links… As a consequence, the amplitude of probability $t_{mn}$ of scattering between two modes $m$ and $n$ will take random values with an amplitude smaller than 1 (generally much smaller when there are many modes). Since these coefficients are chosen randomly in our approach (while preserving the unitarity of the scattering matrix), we do not neglect potential scattering between modes, we simply do not specify the origin of this scattering.
In current experiments, most quantum conductors used to fabricate Josephson junctions are not sensitive to localization on the scale of interest (the superconducting coherence length) for the formation of Andreev molecules. We believe it is irrelevant for the physics we describe here. In presence of localization, the Josephson effects would be suppressed in both junctions and this would be similar to the case of two tunnel junctions. This case is treated in the appendix of our previous work (Ref. 5).
Forming a superconducting loop around a quantum conductor and controlling the magnetic flux through this loop is actually the most common approach to adjust the phase across a Josephson junction. See for example, the experimental work of Pillet et al.
In ref 5, we propose a setup with such a double loop in order to detect an Andreev molecule with a Josephson spectroscopy (fig 4a, attached),
The referee may be confusing phase-bias with voltage-bias.
Attachment:

---

## Round 2 · Referee Report · Anonymous (Referee 4) · 2020-5-19

Report
The manuscript describes a theoretical study of the Andreev spectrum of two superconducting weak links in series, assuming many channels in the links. The work builds up on a previous publication by the same authors, wherein they study the formation of Andreev “molecules” in the single-channel regime, when the weak links are placed sufficiently close (on the scale of the coherence length). They study and interpret the effects of the coherent coupling of the Andreev bound states as a function of the weak link separation, by means of a generalisation of the Beenakker equation.
Let me first provide my own assessment, and then comment on the previous referee reports.
The manuscript is clearly written and pedagogically structured. While the study may not be the most original and groundbreaking in terms of methods and formalism, I believe nonetheless that the results are of interest and importance to the community, and should be published. In particular, since their results are formulated in the scattering matrix language, it seems plausible that their results can be readily “ported” to other important systems, such as Majorana-based junctions. Perhaps, they could highlight a potential connection in their introduction.
Apart from that I think that the manuscript has improved upon resubmission, providing satisfactory responses to the first review round (with reports submitted on March 10 and March 20, respectively).
There remains one particularly critical report (submitted on April 7) wherein the referee doubts that the authors properly describe the multi-channel physics of the junctions (allegedly neglecting the strongly varying tunnelling probability distribution, and disregarding localization effects). Finally, the referee does not seem to understand how the phase bias setup the authors propose (with essentially three contacts, giving rise to two independent phase differences) could possibly work.
The authors response (submitted on April 17) seems convincing to me.
They point out that picking the scattering matrix from a random orthogonal ensemble allows for studying a most generic system, giving naturally rise to strongly varying transmission probability distributions. This is indeed a standard way to describe generic scattering regions. One could potentially criticize that while this method is generic, it may not necessarily help an experimenter to understand his data for a specific realisation - but it seems completely legitimate to me to postpone a more detailed study with more realistic models to subsequent research, e.g., by means of a microscopic description of the scatterers.
As for the localization issue, I believe and support the authors’ assertion that it is irrelevant for the systems they consider, where the conductor forming the junction is sufficiently short.
Finally, the authors explain how to control the phase differences across the three terminals by means of loops enclosing a flux. This is a very standard way of phase control in superconducting circuits, and I do not see any problem with this proposition.
Let me first provide my own assessment, and then comment on the previous referee reports.
The manuscript is clearly written and pedagogically structured. While the study may not be the most original and groundbreaking in terms of methods and formalism, I believe nonetheless that the results are of interest and importance to the community, and should be published. In particular, since their results are formulated in the scattering matrix language, it seems plausible that their results can be readily “ported” to other important systems, such as Majorana-based junctions. Perhaps, they could highlight a potential connection in their introduction.
Apart from that I think that the manuscript has improved upon resubmission, providing satisfactory responses to the first review round (with reports submitted on March 10 and March 20, respectively).
There remains one particularly critical report (submitted on April 7) wherein the referee doubts that the authors properly describe the multi-channel physics of the junctions (allegedly neglecting the strongly varying tunnelling probability distribution, and disregarding localization effects). Finally, the referee does not seem to understand how the phase bias setup the authors propose (with essentially three contacts, giving rise to two independent phase differences) could possibly work.
The authors response (submitted on April 17) seems convincing to me.
They point out that picking the scattering matrix from a random orthogonal ensemble allows for studying a most generic system, giving naturally rise to strongly varying transmission probability distributions. This is indeed a standard way to describe generic scattering regions. One could potentially criticize that while this method is generic, it may not necessarily help an experimenter to understand his data for a specific realisation - but it seems completely legitimate to me to postpone a more detailed study with more realistic models to subsequent research, e.g., by means of a microscopic description of the scatterers.
As for the localization issue, I believe and support the authors’ assertion that it is irrelevant for the systems they consider, where the conductor forming the junction is sufficiently short.
Finally, the authors explain how to control the phase differences across the three terminals by means of loops enclosing a flux. This is a very standard way of phase control in superconducting circuits, and I do not see any problem with this proposition.

---

## Round 2 · Author Response

We have revised our manuscript in order to improve clarity and address the points raised in the two referee reports.
We thank both referees for their comments.
Report 1 suggests that the novelty and significance of our work was undermined by the prevalence in our field of the scattering approach and crossed Andreev reflection (CAR).
The referee should note that it is only recently that one finds an explicit and complete expression of the scattering matrix for a superconductor of finite thickness (see for example an initial version of our previous work arxiv:1809.11011v1, or, in 2019, PRR 1 033212 and arxiv:1912.10307).
More importantly the major novelty of our work lies in the following two conclusions which are relevant for designing or understanding experiments:
a) In Figure 2 and accompanying discussion, we show that having multiple channels does not change qualitatively the signatures of hybridization of Andreev Bound States (ABS).
As mentioned in Report 2, "In the context of quantum information, new quantum circuits including hybrid systems (here with superconducting and normal materials) and various geometries are widely investigated."
Our results help direct experimental efforts using such systems.
b) In Figure 3 and accompanying discussion we show that the scattering amplitudes at maximal hybridization correspond to two specific microscopic mechanisms, double elastic cotunneling and double CAR (Fig. 3 and discussion).
We establish quantitatively the relationship between the well known phenomenon of CAR and elastic cotunneling (NSN systems) to the microscopic processes occurring in our SNSNS system, allowing a deeper, intuitive understanding of the Andreev molecule.
Our manuscript clearly states these two points, both in the abstract and in the conclusion.
We agree with Report 2 that although our results may not be ground-breaking, they "are of real interest for the community".
We hope that referee 1 will also be convinced of the novelty and significance of our manuscript.
We address the minor technical comments of both referees and the accompanying changes to the manuscript in detail below.
We thank both referees for their comments.
Report 1 suggests that the novelty and significance of our work was undermined by the prevalence in our field of the scattering approach and crossed Andreev reflection (CAR).
The referee should note that it is only recently that one finds an explicit and complete expression of the scattering matrix for a superconductor of finite thickness (see for example an initial version of our previous work arxiv:1809.11011v1, or, in 2019, PRR 1 033212 and arxiv:1912.10307).
More importantly the major novelty of our work lies in the following two conclusions which are relevant for designing or understanding experiments:
a) In Figure 2 and accompanying discussion, we show that having multiple channels does not change qualitatively the signatures of hybridization of Andreev Bound States (ABS).
As mentioned in Report 2, "In the context of quantum information, new quantum circuits including hybrid systems (here with superconducting and normal materials) and various geometries are widely investigated."
Our results help direct experimental efforts using such systems.
b) In Figure 3 and accompanying discussion we show that the scattering amplitudes at maximal hybridization correspond to two specific microscopic mechanisms, double elastic cotunneling and double CAR (Fig. 3 and discussion).
We establish quantitatively the relationship between the well known phenomenon of CAR and elastic cotunneling (NSN systems) to the microscopic processes occurring in our SNSNS system, allowing a deeper, intuitive understanding of the Andreev molecule.
Our manuscript clearly states these two points, both in the abstract and in the conclusion.
We agree with Report 2 that although our results may not be ground-breaking, they "are of real interest for the community".
We hope that referee 1 will also be convinced of the novelty and significance of our manuscript.
We address the minor technical comments of both referees and the accompanying changes to the manuscript in detail below.

---

## Round 2 · List of Changes

** Report 1, Issue 1
/The central region of the superconducting wire (the "Andreev molecule") is grounded, and the authors say that this allows them to fix the superconducting phase of the central region at 0. Here they are confusing voltage bias and flux bias. "Grounding" means that the voltage is zero, it does not mean that the phase is zero. The phase difference phi_L-phi_R between the outer ends of the wire can be fixed by a flux bias, but the phase of the central region should then be determined selfconsistently, it is not fixed by the voltage ground./
** Response to Report 1, Issue 1:
The ground has nothing to do with fixing the superconducting phase of the central superconductor. The central superconducting part is grounded in order to allow the left and right junction to carry different supercurrent while preserving current conservation. In practice, this allows to independently current bias the left and right junction in order to measure their respective critical currents, or alternatively to current bias one of the junction while the other is closed by a superconducting loop such that it can be phase biased with a magnetic flux. These two setups allow to detect the non-local nature of ABS in the Andreev molecule (see our first manuscript arXiv:1809.11011).
In our work, the phase of the central superconductor is arbitrarily set to zero. By gauge invariance, this does not imply any loss of generality. What matters physically are the phase differences between superconductors. Since the central phase is chosen to be zero, these phase differences are given by phi_L and phi_R.
We believe the confusion could come from the original phrasing in the caption of Fig. 1:
“The ground connection allows applying the phase differences phi_L,R independently”
What we meant here is that the current can be different in each junction since there is a ground connection in the middle. As a consequence, phi_L and phi_R can be set independently.
We have changed this sentence to:
“The ground connection allows applying the phase differences phi_L,R independently by flowing different current through each junctions.”
The confusion could also come from a similar sentence in the original text, page 5:
“Because of the ground connection, there are effectively two loops connecting the left and right superconductors which allow tuning phi_L,R independently with external magnetic fields.”
This is indeed confusing since the ground is actually not necessary to form loops between superconductors. We have modified this to:
“Each junction can be shorted by a superconducting loop which allow tuning phi_L,R independently with external magnetic fields.”
** Report 1, Issues 2 and 3:
/The authors assume that the electron and hole propagate through the central region with longitudinal momentum k_e,h = k_F ± i/ξ. In a multi-mode junction the longitudinal momentum can vary between 0 and the maximal value of k_F. Setting it equal to k_F for all modes does not seem justified./
/On page 6 the authors write "we ignore fast phase oscillations in t_S and r_S arising from the small Fermi wavelength by fixing k_F l arbitrarily and independent of l". This assumption makes no sense to me. In a phase coherent treatment these phase oscillations should play a crucial role./
** Response to Report 1, Issues 2 and 3:
It is entirely correct that in general the longitudinal part of the wavevector can be different for each channel. In Fig. 2, we have chosen $k_F l\pmod{2\pi} = 0$ for simplicity and convenience. This implies that the phase acquired due to propagation only is the same for each channel. There is no physical reason to make this particular choice. In an actual experiment, these phases would actually take random values between 0 and 2pi for each channel. This is something we can easily introduce in our calculations but it does not change qualitatively the shape of our spectra (the main difference would be a variation of the amplitude of the ABS avoided crossings). Moreover, since the normal scattering matrices S_L and S_R are chosen randomly, they already introduce different scattering phases, which prevents peculiar situations where all channels would host constructive or destructive interference maximizing or minimizing the magnitude of each avoided crossings.
We have modified the text at the bottom of page 5 to state, "the longitudinal part of the wavevector can take any value between 0 and $k_F$." and added the following to the next paragraph:
"For convenience and visibility we set $k_Fl$ to constant values in the scattering coefficients while maintaining $k_F l \gg 1$.
In principle each channel may have a different phase factor resulting from interference but such offsets are already included via the random unitary scattering matrices $S_{L,R}$ and do not change the results qualitatively."
** Report 1, Issue 4
/The plots are calculated by "randomly generated symmetric unitary matrices S_L and S_R". These matrices should represent the weak link, which I presume is a tunnel junction. Since no disorder is included in the superconductor, the randomness needed for this assumption is not present and I do not understand the justification for this choice./
** Response to Report 1, Issue 4
The randomly generated scattering matrices S_L and S_R indeed represent the weak link. The weak link is not necessarily tunnel junctions (for which the non-local Josephson Effect is weak) but could be any quantum conductor. Since our approach is 1D, our model preferably describes one-dimensional quantum conductors, such as carbon nanotubes or semiconducting nanowires, but in principle S_L and S_R could also describe any quantum conductors in 2 or 3 dimensions.
The reason we choose random S_L and S_R for quantum conductors with many conduction channels (Fig. 2) is not to include disorder but rather to keep a very generic approach with no assumption made on the nature of the conduction channel. For example, they have no reason to be the same than in the central superconductor.
The disorder of the superconducting region is not included in the matrix S_S. It could perfectly be taken into account in our approach by adding in series with S_S two normal scattering matrices S_S^L and S_S^R, on its left and on its right. They would describe scattering at the left and right interfaces of the superconductor and disorder in between. Nevertheless, it is possible to combine S_S^L and S_L (or S_S^R and S_R) into a single scattering matrix describing the left (resp. right) quantum conductor, which would be formally equivalent to what is done in our manuscript.
A better description of disorder could be done using a tight-binding approach as was done in the initial version of our previous work (arxiv:1809.11011v1) or using a semi-classical approach in more recent works by the Nazarov group. These results shows that there is no qualitative change in the formation of an Andreev molecule when one introduces disorder. Disorder mainly leads to a weakening of the ABS hybridization because of a reduction of the superconducting length xi (xi is replace by sqrt[xi*mfp] where mfp is the mean free path). However, it should remain measurable as long as the junction are separated by a distance comparable to this new xi.
To clarify how disorder at the superconductor or weak link-superconductor interface could be incorporated into S_L and S_R, we have added the following sentence at the bottom of page 2, "These matrices can in principle include additional scattering at the superconductor-weak link interface."
** Report 2, Issue 1
/The authors present results for a fixed number of channels (N = 20). It would have been interesting to discuss the effect of the number of channels in more details. This can be of real use for practical realizations with gated semiconducting nanowires./
** Response to Report 2, Issue 1
We agree that our results are of relevance to experiments with gated semiconducting nanowires and state this in the manuscript.
In experiments with such nanowires the number of highly transmitting channels is often in the single digits.
In our calculations with random scattering matrices we chose N = 20 in order to ensure that we would have a comparable number of highly transmitting channels, as shown in Figure 2.
In addition, for larger N, visibility in the spectra is reduced.
To clarify that there is no qualitative difference in the spectra with increasing N, we have added the following sentence at the end of section 3:
"Even though the spectra will become more dense as the number of channels is increased, this symmetry breaking will be relevant experimentally as long as $l\lesssim\xi_0$."
** Report 2, Issue 2
/The origin of hybridization is due to (inverse) proximity effect in the central superconductor and explains the L/xi decay dependence. This inverse proximity effect is known to be strong when the transparency at the S/N interface is good (close to 1). But at the same time, the superconducting gap is locally reduced. On the contrary, when the transparency is smaller, the superconducting gap is restored but (inverse) proximity effect is less efficient./
** Response to Report 2, Issue 2
For semiconducting weak links in which the carrier density is much lower than in the metallic electrodes, the inverse proximity effect is negligible.
However it is true that the quality of the superconductor-semiconductor interface may play a strong role in hybridization.
In our formalism it is possible to add additional scattering at this interface by incorporating it into the matrices S_L and S_R, and as mentioned above, we have modified the text accordingly: "These matrices can in principle include additional scattering at the superconductor-weak link interface."
Although a more realistic 3D treatment of this interface is out of the scope of our manuscript, we provide references to recent preprints by Nazarov et al which address this question.
/The central region of the superconducting wire (the "Andreev molecule") is grounded, and the authors say that this allows them to fix the superconducting phase of the central region at 0. Here they are confusing voltage bias and flux bias. "Grounding" means that the voltage is zero, it does not mean that the phase is zero. The phase difference phi_L-phi_R between the outer ends of the wire can be fixed by a flux bias, but the phase of the central region should then be determined selfconsistently, it is not fixed by the voltage ground./
** Response to Report 1, Issue 1:
The ground has nothing to do with fixing the superconducting phase of the central superconductor. The central superconducting part is grounded in order to allow the left and right junction to carry different supercurrent while preserving current conservation. In practice, this allows to independently current bias the left and right junction in order to measure their respective critical currents, or alternatively to current bias one of the junction while the other is closed by a superconducting loop such that it can be phase biased with a magnetic flux. These two setups allow to detect the non-local nature of ABS in the Andreev molecule (see our first manuscript arXiv:1809.11011).
In our work, the phase of the central superconductor is arbitrarily set to zero. By gauge invariance, this does not imply any loss of generality. What matters physically are the phase differences between superconductors. Since the central phase is chosen to be zero, these phase differences are given by phi_L and phi_R.
We believe the confusion could come from the original phrasing in the caption of Fig. 1:
“The ground connection allows applying the phase differences phi_L,R independently”
What we meant here is that the current can be different in each junction since there is a ground connection in the middle. As a consequence, phi_L and phi_R can be set independently.
We have changed this sentence to:
“The ground connection allows applying the phase differences phi_L,R independently by flowing different current through each junctions.”
The confusion could also come from a similar sentence in the original text, page 5:
“Because of the ground connection, there are effectively two loops connecting the left and right superconductors which allow tuning phi_L,R independently with external magnetic fields.”
This is indeed confusing since the ground is actually not necessary to form loops between superconductors. We have modified this to:
“Each junction can be shorted by a superconducting loop which allow tuning phi_L,R independently with external magnetic fields.”
** Report 1, Issues 2 and 3:
/The authors assume that the electron and hole propagate through the central region with longitudinal momentum k_e,h = k_F ± i/ξ. In a multi-mode junction the longitudinal momentum can vary between 0 and the maximal value of k_F. Setting it equal to k_F for all modes does not seem justified./
/On page 6 the authors write "we ignore fast phase oscillations in t_S and r_S arising from the small Fermi wavelength by fixing k_F l arbitrarily and independent of l". This assumption makes no sense to me. In a phase coherent treatment these phase oscillations should play a crucial role./
** Response to Report 1, Issues 2 and 3:
It is entirely correct that in general the longitudinal part of the wavevector can be different for each channel. In Fig. 2, we have chosen $k_F l\pmod{2\pi} = 0$ for simplicity and convenience. This implies that the phase acquired due to propagation only is the same for each channel. There is no physical reason to make this particular choice. In an actual experiment, these phases would actually take random values between 0 and 2pi for each channel. This is something we can easily introduce in our calculations but it does not change qualitatively the shape of our spectra (the main difference would be a variation of the amplitude of the ABS avoided crossings). Moreover, since the normal scattering matrices S_L and S_R are chosen randomly, they already introduce different scattering phases, which prevents peculiar situations where all channels would host constructive or destructive interference maximizing or minimizing the magnitude of each avoided crossings.
We have modified the text at the bottom of page 5 to state, "the longitudinal part of the wavevector can take any value between 0 and $k_F$." and added the following to the next paragraph:
"For convenience and visibility we set $k_Fl$ to constant values in the scattering coefficients while maintaining $k_F l \gg 1$.
In principle each channel may have a different phase factor resulting from interference but such offsets are already included via the random unitary scattering matrices $S_{L,R}$ and do not change the results qualitatively."
** Report 1, Issue 4
/The plots are calculated by "randomly generated symmetric unitary matrices S_L and S_R". These matrices should represent the weak link, which I presume is a tunnel junction. Since no disorder is included in the superconductor, the randomness needed for this assumption is not present and I do not understand the justification for this choice./
** Response to Report 1, Issue 4
The randomly generated scattering matrices S_L and S_R indeed represent the weak link. The weak link is not necessarily tunnel junctions (for which the non-local Josephson Effect is weak) but could be any quantum conductor. Since our approach is 1D, our model preferably describes one-dimensional quantum conductors, such as carbon nanotubes or semiconducting nanowires, but in principle S_L and S_R could also describe any quantum conductors in 2 or 3 dimensions.
The reason we choose random S_L and S_R for quantum conductors with many conduction channels (Fig. 2) is not to include disorder but rather to keep a very generic approach with no assumption made on the nature of the conduction channel. For example, they have no reason to be the same than in the central superconductor.
The disorder of the superconducting region is not included in the matrix S_S. It could perfectly be taken into account in our approach by adding in series with S_S two normal scattering matrices S_S^L and S_S^R, on its left and on its right. They would describe scattering at the left and right interfaces of the superconductor and disorder in between. Nevertheless, it is possible to combine S_S^L and S_L (or S_S^R and S_R) into a single scattering matrix describing the left (resp. right) quantum conductor, which would be formally equivalent to what is done in our manuscript.
A better description of disorder could be done using a tight-binding approach as was done in the initial version of our previous work (arxiv:1809.11011v1) or using a semi-classical approach in more recent works by the Nazarov group. These results shows that there is no qualitative change in the formation of an Andreev molecule when one introduces disorder. Disorder mainly leads to a weakening of the ABS hybridization because of a reduction of the superconducting length xi (xi is replace by sqrt[xi*mfp] where mfp is the mean free path). However, it should remain measurable as long as the junction are separated by a distance comparable to this new xi.
To clarify how disorder at the superconductor or weak link-superconductor interface could be incorporated into S_L and S_R, we have added the following sentence at the bottom of page 2, "These matrices can in principle include additional scattering at the superconductor-weak link interface."
** Report 2, Issue 1
/The authors present results for a fixed number of channels (N = 20). It would have been interesting to discuss the effect of the number of channels in more details. This can be of real use for practical realizations with gated semiconducting nanowires./
** Response to Report 2, Issue 1
We agree that our results are of relevance to experiments with gated semiconducting nanowires and state this in the manuscript.
In experiments with such nanowires the number of highly transmitting channels is often in the single digits.
In our calculations with random scattering matrices we chose N = 20 in order to ensure that we would have a comparable number of highly transmitting channels, as shown in Figure 2.
In addition, for larger N, visibility in the spectra is reduced.
To clarify that there is no qualitative difference in the spectra with increasing N, we have added the following sentence at the end of section 3:
"Even though the spectra will become more dense as the number of channels is increased, this symmetry breaking will be relevant experimentally as long as $l\lesssim\xi_0$."
** Report 2, Issue 2
/The origin of hybridization is due to (inverse) proximity effect in the central superconductor and explains the L/xi decay dependence. This inverse proximity effect is known to be strong when the transparency at the S/N interface is good (close to 1). But at the same time, the superconducting gap is locally reduced. On the contrary, when the transparency is smaller, the superconducting gap is restored but (inverse) proximity effect is less efficient./
** Response to Report 2, Issue 2
For semiconducting weak links in which the carrier density is much lower than in the metallic electrodes, the inverse proximity effect is negligible.
However it is true that the quality of the superconductor-semiconductor interface may play a strong role in hybridization.
In our formalism it is possible to add additional scattering at this interface by incorporating it into the matrices S_L and S_R, and as mentioned above, we have modified the text accordingly: "These matrices can in principle include additional scattering at the superconductor-weak link interface."
Although a more realistic 3D treatment of this interface is out of the scope of our manuscript, we provide references to recent preprints by Nazarov et al which address this question.

---

## Editorial Decision

published